# Quality of care offered by health care retail markets for medication abortion self-management: Findings from states in Nigeria and India

Mridula Shankar[1¤]*, Elizabeth Omoluabi[2,3], Funmilola M. OlaOlorun[4], Anoop Khanna[5], Danish Ahmad[5], Caroline Moreau[1,6], Suzanne O. Bell[1]

**1** Department of Population, Family and Reproductive Health, Johns Hopkins Bloomberg School of Public Health, Baltimore, Maryland, United States of America, **2** Department of Statistics and Population Studies, University of the Western Cape, Cape Town, South Africa, **3** Centre for Research Evaluation Resources and Development, Nigeria, **4** Department of Community Medicine, College of Medicine, University of Ibadan, Ibadan, Oyo, Nigeria, **5** Indian Institute of Health Management Research, Jaipur, India, **6** Soins Primaires et Prévention, CESP Centre for Research in Epidemiology and Population Health, U1018, Inserm, Villejuif, France

¤ Current address: Gender and Women's Health Unit, Nossal Institute for Global Health, Melbourne School of Population and Global Health, Carlton, Victoria, Australia
* mridula.shankar@unimelb.edu.au

**Data Availability Statement:** The data underlying this article cannot be shared publicly due to the

## Abstract

Dispensing of misoprostol and mifepristone by pharmacies and chemist shops for self-management of medication abortion (MA) fills a crucial gap in settings where abortion care by trained health professionals is not readily available. This promising service delivery pathway, endorsed by the World Health Organization (WHO), is hindered by concerns of poor-quality care. Simulated clients collected data on MA pill dispensing practices from 92 pharmacies and chemist shops in three Nigerian states and 127 pharmacies in an Indian state that we have anonymized. Guided by the WHO's abortion guideline, we measured process-related quality indicators such as medication use instructions, warning signs, and respectful treatment among other aspects. We aggregated indicators under three domains: technical competence, information given to clients, and client experience. Overall, 51% of facilities in the Nigerian states and 32% in the Indian state offered MA pills. Most dispensing facilities offered the misoprostol-only regimen in Nigeria (68%) and the combination regimen in the Indian state (83%). Among facilities offering MA pills, 26% in Nigeria and 78% in the Indian state provided correct instructions on route of pill administration. Accurate information on the appropriate interval between pill type in the combination regimen was low in Nigeria (27%) and the Indian state (14%). Excessive bleeding as a warning sign was discussed more frequently in the Indian (56%) versus Nigerian states (32%); other abnormal bleeding patterns were rarely mentioned. Aggregate technical competency scores were low at 18% in Nigeria and 34% in the Indian state, with highest scores for client experience at 90% and 91% respectively. Findings suggest that people using MA pills purchased from the retail market are not given accurate and adequate information for most effective self-use. If MA

sensitivity of the topic and to protect the privacy of individuals that participated in the study.

**Funding:** This study was funded by an Anonymous Donor (grant number 127941) and the Bill & Melinda Gates Foundation (grant number OPP10709004, PI not a co-author) via grants to Performance Monitoring for Action (https://www.pmadata.org) at the Johns Hopkins Bloomberg School of Public Health. The funders had no role in study design, data collection and analysis, decision to publish, or preparation of the manuscript.

**Competing interests:** The authors have declared that no competing interests exist.

self-management remains outside regulatory boundaries, technical quality will remain substandard, imposing unnecessary costs to people, their health, and health systems.

## Introduction

Yearly, four in 100 reproductive-aged women undergo an abortion [1]. In countries where legal provision of abortion is restricted to narrow grounds and/or where health systems are poorly resourced, peoples' reliance on clandestine abortion has motivated a longstanding public health focus on abortion safety [2]. Safety estimates suggest that 45% of abortions globally are not safe, and the risk of unsafe abortions is disproportionately experienced by women in low-and middle-income regions [3]. In recent decades the abortion landscape has evolved to include models of self-management using misoprostol with or without mifepristone, both of which are World Health Organization (WHO)-recommended medications for medication abortion (MA) and on the model list of essential medicines for health systems [4, 5]. Self-management of MA encompasses several service-delivery models that offer the individual an active and autonomous role in inducing and managing their abortion, in contrast to a clinician-performed procedure on the recipient of care. In this paper, we concern ourselves with a form of abortion self-management that is more typical of legally restrictive and/or poorly resourced public health care settings, where misoprostol and mifepristone (when available), and hereafter referred to as MA pills, are purchased with neither prescriptions nor access to high quality information from pharmacies, chemist shops and other medicine vendors, and used for self-inducing an abortion. We want to distinguish this form of self-management from those models that are accompanied by support and trained guidance on appropriate use of the medication regimen(s), which have demonstrated high levels of safety and efficacy [6].

Unguided self-management with MA pills has significantly reduced the health risks previously associated with clandestine abortions outside clinical settings, as evidenced by decreased severe abortion-related morbidity and mortality [7]. Recent studies in Lagos state and Cambodia that prospectively followed women who underwent self-induced abortions using MA pills sourced from drug sellers and pharmacies have also documented high completion rates without additional intervention [8, 9]. Yet, research in Asian, Latin American and sub-Saharan African countries suggests that information provided during pill purchases is often inadequate or incorrect [10–13], with dosages provided not meeting technical recommendations [9, 14–16]. In Mexico and Colombia, alongside declines in severe abortion-related morbidity and mortality, there are indications of increased post-abortion care caseloads [17, 18]. In India, where most abortions are self-managed, the rate of accompanying complications is unusually high [19], even when accounting for the fact that some women seek facility-based follow-up care in the absence of true medical complications [19, 20]. Collectively, these data present a mixed picture but suggest that when correct protocols for effective and safe use of MA pills are not being followed in informal use contexts, some women are placed at unnecessary risk for preventable complications.

With this reality, an expanded focus on quality—which the WHO defines as care that is *safe, effective, timely, efficient, equitable and people-centred*—can serve as an integrating framework to appraise multiple components of abortion care that are central to its experience and amenable to improvements [21]. A quality-focused lens applied to abortion can expand measurement beyond safety to assess technically competent care and client experience. The latter is especially relevant as people's decisions and experiences of abortion are closely linked to

concerns around privacy and respectful treatment [22]. It can help re-frame abortion as a legitimate reproductive health care service that is deserving of high-quality care, encouraging policy and programmatic uptake of empirical data to inform service improvements. Incorporated into its definition is the often-neglected dimension of equity that can motivate disaggregation of data to identify and prioritize the needs of disadvantaged groups who bear a disproportionate burden of adverse outcomes. Finally, in the context of recent normative guidance on integrating self-care interventions for sexual and reproductive health—including abortion—into wider health care reforms [23], and the updated 2022 abortion care guideline that has, for the first time, a service delivery recommendation that pharmacy and pharmacy workers perform tasks for medical management of abortion [5], the quality of service-delivery accompanying the provision of MA pills for abortion self-management cannot be overlooked.

Indicators focused on quality have largely been developed to assess facility-based abortion care. Despite growing acknowledgment of the integral role played by the informal health care sector in care provision [24], including in the case of abortion [25], systematic efforts to widen the scope of quality measurement to include informal providers has only been a recent endeavor [11, 12, 26, 27]. In this study, we utilize a simulated client approach [28] to operationalize quality measurement associated with the informal dispensing of MA pills through pharmacies and chemist shops in three states in Nigeria and one state in India. We chose this approach due to concerns that dispensing behaviors are likely to be under or mis-reported in survey-based measures. These sites were part of a larger abortion measurement project and were selected for this study due to an established (India) and growing (Nigeria) reliance on informally sourced MA pills for self-managed abortions. Both geographical settings differ in the cultural and legal contexts surrounding abortion, which could hold relevance in operationalizing quality and its measurement, motivating this cross-country analysis.

This study has two aims: to define, measure and aggregate process indicators of quality that measure technical competency and client experience associated with informal dispensing of MA pills, and to assess disparities in the availability of MA pills in the informal sector by socio-demographic characteristics of reproductive-aged women living in communities where these sampled drug vendors are located.

## Methods

### Study contexts

Both abortion laws and the availability of safe abortion services differ in the two study settings. The legal framework in Nigeria only permits abortion if continuing the pregnancy puts a woman's life at risk. Nevertheless, abortion is common (33 per 1000 reproductive-aged women [29]), clandestine, and associated safety remains poor [30]. Misoprostol is registered in Nigeria for the treatment of gastric ulcers and postpartum hemorrhage and can be bought from a pharmacy with an accompanying prescription for use to treat these on-label indications. Mifepristone was registered in 2017 and has been made available in a combination regimen. Data on the availability and use of MA pills in Nigeria are sparse; a recent systematic review suggests that availability and diffusion of these medicines is low as indicated by the continued high rates of unsafe abortion-related maternal morbidity and mortality [31]. However, a review of referral hospital medical record data from 2006–14 demonstrated a trend of growing misoprostol use, associated with less severe morbidity [32]. Researchers have also recently documented informal misoprostol sales through pharmacies and drug vendors in Lagos state [9, 26].

In India, the law is more permissive, allowing women to terminate pregnancies for socio-economic reasons in addition to medical risks. The abortion rate is 48 per 1000 reproductive-

aged women [29], yet abortion service provision within the public sector is limited, particularly in primary care facilities [33, 34]. Estimates from 2015 indicate that nearly three-quarters of abortions in India were performed using MA pills supplied outside of legally regulated health care facilities [35]. Literature on the quality of care associated with this form of provision is limited, though existing data suggest that it is sub-standard [36, 37].

### Data sources and sampling

This analysis uses data from a simulated client study conducted amongst urban pharmacies and chemist shops in three Nigerian states and one state in India. We have anonymized study locations in consultation with ethics committees to protect research participants from any potential repercussions due to the topic's sensitivity. We draw on two additional data sources: (1) survey data from these pharmacies and chemist shops and (2) survey data from reproductive-aged women 15–49 years of age residing in the geographical areas where these pharmacies and chemist shops were located at the time of the study. These data were collected by Performance Monitoring for Action (formerly Performance Monitoring and Accountability 2020), a multi-country project that conducts female and service delivery point (SDP) surveys to monitor a range of reproductive health-related indicators [38]. In each round of data collection, trained female interviewers administered face-to-face surveys with reproductive-aged women and the most knowledgeable staff at sampled SDPs using mobile phone-based software to record survey responses. Investigators added an abortion module to the 2018 female and SDP surveys in Nigeria and the Indian state. We subsequently included a simulated client study amongst a subset of pharmacies and chemist shops that completed the SDP survey, to measure MA pill availability and the quality of care associated with informal dispensing of these medicines.

The sampling design for the simulated client study was linked to the SDP survey, which in turn was linked to the design of the female survey. National and state-representative samples of reproductive-aged women were generated using a multi-stage urban/rural stratified cluster sampling approach. In the penultimate stage, enumeration areas (EA), which are geographical units encompassing approximately 200 households, were selected using probability proportional to size sampling, followed by a random sampling of households in each selected EA. All women aged 15–49 who lived in a selected household or spent the previous night in that household were eligible to participate in the female survey after providing informed consent. Data collection for the female surveys occurred from April through May 2018 in the three Nigerian states, and from April through June 2018 in the Indian state.

The SDP sample was composed of public and private facilities that served the population of reproductive-aged women in the selected EAs. Up to three public facilities at the primary, secondary and tertiary levels of care whose catchment areas contained the EAs were included. Separately, interviewers mapped each EA to locate all private SDPs within its boundaries. If there were three or fewer private SDPs per EA, interviewers surveyed each, otherwise supervisors randomly selected three to be surveyed. Included in this sample of private facilities were pharmacies and chemist shops (also called patent medicine vendors in Nigeria), both of which sell pharmaceutical pills, however, chemist shops are typically unlicensed, do not usually employ a pharmacist, and have restrictions on pills that can be sold. Even in facilities that are owned or run by pharmacists, the day-to-day operations and dispensing of medications are usually conducted by staff whose educational qualifications and training vary. We use survey and simulated client data from pharmacy and chemist shop SDPs and do not distinguish between them in the analysis.

A total of 260 pharmacies and chemists completed the 2018 SDP surveys in the three Nigerian states between April and May 2018 and 242 completed surveys in the Indian state between April and June 2018. From this sampling frame, we restricted eligibility for the simulated client study to 93 SDPs in Nigeria and 131 SDPs in the Indian state based on their urban locations to maximize the anonymity and safety of simulated clients and research participants (SDP staff). In rural or isolated areas, an outsider may have been more noticeable. Data collection for the simulated client study took place from July through August 2018 in all locations.

## Training and data collection

We selected male and female simulated clients after careful consideration of their ability to match profiles of typical clients seeking abortion pills for themselves or on behalf of another. They participated in a four-day in-person training program at each site to fine-tune the study design and protocol, finalize simulated client profile descriptions and the standardized set of probes and responses, and discuss ethical considerations and safety protocols. Data collection activities were piloted and minorly adjusted.

Study procedures differed slightly between the two settings. In Nigeria, each sampled pharmacy and chemist shop was visited twice, once by a male and once by a female simulated client to determine if staff/client interactions were impacted by client gender, with visits roughly two weeks apart. In the Indian state, male simulated clients were randomly assigned to one half of the selected facilities and female simulated clients were assigned to the other half as in-country investigators were concerned that dual visits to each facility within a short duration may arouse suspicion. The final male and female client profiles (one for each gender) were contextualized for each setting (S1 Table). In Nigeria, male simulated clients presented seeking abortifacients on behalf of their girlfriends, whereas female simulated clients did so for themselves. In the Indian state, male simulated clients sought abortifacients on behalf of their wives, and female simulated clients on behalf of their younger married sisters-in-law. In both settings, simulated clients did not present prescriptions for abortifacient pills. The gestational age of the pregnancy was reported as six weeks in all instances. Since the interaction was also designed to gather information on staff knowledge of appropriate use of MA pills, simulated clients offered these pills were trained to ask probing questions about directions for use, side-effects, potential complications, and locations for follow-up care if information was not offered unprompted by staff. Interactions were terminated prior to the purchase of any pills in accordance with ethical approvals given that such prescription-less purchases would have been illegal. After each visit, simulated clients completed a self-administered electronic questionnaire on their mobile phone designed to capture details of their interactions with pharmacy staff. No information that could identify the staff member was recorded. For each question on staff knowledge, simulated clients also recorded whether the information given was prompted by the client or provided voluntarily. Staff were unaware of study participation to ensure that we captured typical client interactions.

## Ethics approvals

We received prospective ethical approvals for all study components as follows: (1) The Johns Hopkins Bloomberg School of Public Health (JHBSPH) provided approval for the female surveys with reproductive aged women and the simulated client study with pharmacies and chemist shops. The JHSBPH IRB deemed the SDP surveys as non-human subjects research since facility data collected on staffing, stock of medications, services provided etc., are unrelated to personal information, perspectives, or opinions. (2) The National Health Research Ethics Committee of Nigeria, and the Institutional Ethics Committee at our collaborating

institution in the Indian state provided ethics approvals for the female surveys, the SDP surveys, and the simulated client study. (3) Ethics approvals were also obtained for the simulated client study from governmental ministries in the three Nigerian states where this study component was implemented. Additional information regarding the ethical, cultural, and scientific considerations specific to inclusivity in global research is included in the Supporting Information (S1 Checklist).

## Quality of care measures

The process indicators of quality used in this analysis are informed by the WHO 2022 abortion care guideline [5]. Process measures illuminate the extent to which care provided during patient-provider interactions concur with current evidence on best practices, including aspects such as client assessment, diagnostic accuracy, provision of appropriate treatment(s), client counselling, and respectful care. We restricted the measurement of quality to pharmacies and chemist shops that offered MA pills either in the form of the combination regimen (mifepristone and misoprostol) or misoprostol alone. SDPs that offered only non-recommended medicines (Nigeria: n = 9, Indian state: n = 15) were excluded from this analysis on the basis that such drugs are unsafe and/or ineffective and would automatically qualify as poor-quality care. Between thirteen and seventeen (based on regimen offered) indicators captured information under three domains: technical competence, information given to clients and client experience (Table 1).

**Technical competence.** We used two indicators—enquiry about pregnancy confirmation and gestational age—to determine whether staff took preliminary steps to assess eligibility for medication abortion. For the remaining indicators capturing staff knowledge, we used data from <u>all</u> responses, whether prompted by the client or given voluntarily. In the context of a gestational age of six weeks, we assessed whether the directions for use of the MA pills were accurate. For the combination drug regimen, this would include instructions to ingest mifepristone orally, followed by vaginal, buccal or sublingual ingestion of misoprostol, with a 24–48 hour waiting period between mifepristone and misoprostol. If only misoprostol was offered, we assessed whether staff provided instructions to take misoprostol orally, vaginally, buccally, or sublingually. Next, we determined whether the provider offered accurate information on normal signs to expect (menstrual-like bleeding, uterine cramping) side effects (nausea, vomiting, diarrhea, chills, and fever) and warning signs of complications (prolonged and heavy bleeding, intermittent or no bleeding, chills and fever lasting more than 24 hours, severe abdominal pain lasting more than 24 hours). Finally, we assessed if staff provided information on appropriate locations for follow-up in the event of potential complications. These locations included public or private tertiary, secondary or primary-level facilities, private doctors, or non-governmental organizations.

**Information given to clients.** Several studies have documented the "know-do" gap, i.e., the discrepancy between the technical knowledge that providers have, and the actual care provided to clients under typical conditions [24]. Under this domain, we captured whether staff provided correct information on use of the pills, side-effects, warning signs, and locations for follow-up care to clients voluntarily, i.e., unprompted by the simulated client. Another indicator determined whether the client was offered post-abortion contraception information.

**Client experience.** In the third and final domain, we used three indicators, each measured on a five-part Likert scale (strongly agree to strongly disagree with a neutral option) to determine the client's overall experience: whether s/he was treated respectfully during the interaction, whether s/he experienced any disapproval from staff expressed verbally, and whether s/he experienced any disapproval from staff based on body language.

**Table 1. Construction of domains and indicators for process measures of quality of care informed by WHO's 2022 abortion care guideline.**

| Domains | Variables | Allocated points |
|---|---|---|
| **I. Technical Competence** | Enquiry about pregnancy confirmation | 1 |
| | Enquiry about gestational age or LMP | 1 |
| | Route of drug administration [If combination regimen: oral route for mifepristone and vaginal, buccal or sublingual routes for misoprostol If misoprostol-only regimen: oral, vaginal, buccal or sublingual routes] | 1 |
| | Order of pills[1] [mifepristone followed by misoprostol] | 1 |
| | Interval between mifepristone and misoprostol[1] [24–48 hours] | 1 |
| | Discussed abortion symptoms [bleeding and cramping] | 2 |
| | Discussed abortion side-effects [nausea, diarrhea, vomiting, chills, fever] | 5 |
| | Discussed warning signs [excessive bleeding, intermittent bleeding, no bleeding, severe abdominal pain lasting >24hrs, fever lasting >24hrs] | 5 |
| | Discussed appropriate locations for follow-up care [public and private health care facilities] | 1 |
| Maximum domain score for Technical Competence | Combination regimen | 18 |
| | Misoprostol-only regimen | 16 |
| **II. Information Given to Client** | Unprompted information on route of drug administration | 1 |
| | Unprompted information on order of pills[1] | 1 |
| | Unprompted information on interval between mifepristone and misoprostol[1] | 1 |
| | Unprompted information on abortion symptoms | 2 |
| | Unprompted information on abortion side-effects | 5 |
| | Unprompted information on warning signs | 5 |
| | Unprompted information on locations for follow-up care | 1 |
| | Discussed post-abortion contraception | 1 |
| Maximum domain score for Information Given to Client | Combination regimen | 17 |
| | Misoprostol-only regimen | 15 |
| **III. Interpersonal Relationship** | Treated respectfully during interaction | 1 |
| | Verbal disapproval when asked for pills[2] | 1 |
| | Facial expressions indicated disapproval when asked for pills[2] | 1 |
| Maximum domain score for Interpersonal Relationship | | 3 |

[1] Only for combination regimen

[2] These items were reverse-coded so that a score of 1 indicates better performance

## Quality measurement and scoring

We coded most indicators as binary variables, with a point allocated for each correct action performed (e.g., enquiry about gestational age) or accurate information provided (e.g., correct route for drug administration). For a minority of indicators requiring multiple pieces of information, we allocated a point for each component. For example, a maximum of five points was allocated to the question on side-effects, one point each for nausea, vomiting, diarrhea, chills,

and fever (Table 1). For variables related to client experience, we collapsed the ordinal Likert scale data into binary agree and disagree/neutral variables, combing the latter-two categories due to small numbers.

We summed the indicators within each of the three domains to generate raw total scores, and subsequently converted domain totals into percentages. For instance, a raw score of 5/18 for technical competency equated to a percentage score of 28% for that domain. We chose this approach as it retains all information relevant to the original domain score, does not assume a continuous response distribution, and allows valid comparisons of percentage scores across domains.

In Nigeria, since two simulated clients visited each facility, we used two scoring approaches: First, for facilities that offered MA pills to both clients, we assigned a point for each indicator if accurate information was provided in at least one of the two interactions. In a sensitivity analysis, we assessed the absolute change in the distribution of each quality indicator if we instead assigned a point only when the correct action or information occurred in both interactions. Second, for the facility domain score, we calculated the simple arithmetic mean of the two responses for each indicator within a domain prior to aggregation.

## Analyses

We evaluated the proportion of pharmacies and chemist shops that reported mifepristone and misoprostol availability on the SDP survey overall, and by whether these facilities offered MA pills to simulated clients. We tested for significant differences in reporting versus dispensing of MA pills using the McNemar test for paired nominal data.

To examine availability within a local market, we calculated the proportion of EAs having at least one SDP offering MA pills. Next, we conducted bivariate and multivariable logistic regression analyses to examine any systematic differences in drug availability by the characteristics of women living in the sampled EAs, using a sandwich estimator to account for clustering of responses within an EA.

Among SDPs visited by simulated clients, we assessed differences in dispensing behavior by simulated client gender using chi-square tests for data from the Indian state or McNemar tests in Nigeria where each facility was visited twice. We also determined the proportion of facilities offering non-recommended products in conjunction with or instead of MA pills, and the specific products offered. Among facilities offering MA pills, we calculated the percentage distribution of each quality-of-care indicator within the three quality domains, prior to generating domain specific quality scores overall and disaggregated by client gender. All analyses were conducted using Stata v.15.1 [39].

## Results

### Facility reporting and dispensing characteristics

The simulated client study was implemented in 47 EAs across the three Nigerian states with clients successfully visiting a total of 92 pharmacies and chemist shops (98.9%); in the Indian state, it was implemented in 43 EAs and clients visited 127 pharmacies (96.9%). Among these vendors, 27.5% in Nigeria and 0.8% in the Indian state reported selling any MA pills (misoprostol with or without mifepristone) on the SDP survey (Table 2). In comparison, a significantly higher percentage—just over one-half (51.1%) in the Nigerian states, and nearly one-third (32.3%) in the Indian state—offered MA pills for early pregnancy termination. On this basis, we found that in a majority of sampled EAs—57.3% in the Nigerian states and 55.8% in the Indian state—at least one SDP offered MA pills for early pregnancy termination.

**Table 2. Sample characteristics of facilities visited by simulated clients in three Nigerian states and the Indian state, including by whether or not clients were offered medication abortion pills for early pregnancy termination.**

| | Nigerian states | | | | | | Indian state | | | | | |
| --- | --- | --- | --- | --- | --- | --- | --- | --- | --- | --- | --- | --- |
| | Offered MA pills | | Not offered MA pills | | All facilities | | Offered MA pills | | Not offered MA pills | | All facilities | |
| Characteristic | N | % | N | % | N | % | N | % | N | % | N | % |
| **SDP type** | | | | | | | | | | | | |
| Pharmacy | 7 | 14.9 | 4 | 8.9 | 11 | 12.0 | 41 | 100.0 | 86 | 100.0 | 127 | 100.0 |
| Chemist shop | 40 | 85.1 | 41 | 91.1 | 81 | 88.0 | -- | -- | -- | -- | -- | -- |
| **At least one pharmacist on staff** | 10 | 21.3 | 11 | 24.4 | 21 | 22.8 | 38 | 92.7 | 79 | 91.9 | 117 | 92.1 |
| **Reported medication abortion drug availability on SDP survey** | 19 | 41.3 | 6 | 13.3 | 25 | 27.5 | 0 | 0.0 | 1 | 1.2 | 1 | 0.8 |
| **Offered simulated client(s) any medications for pregnancy termination[1]** | | | | | | | | | | | | |
| None offered | -- | -- | 36 | 80.0 | 36 | 39.1 | -- | -- | 73 | 84.9 | 73 | 57.5 |
| Mifepristone and misoprostol | 6 | 12.8 | -- | -- | 6 | 6.5 | 34 | 82.9 | -- | -- | 34 | 26.8 |
| Misoprostol only | 32 | 68.1 | -- | -- | 32 | 34.8 | 6 | 14.6 | -- | -- | 6 | 4.7 |
| Both regimens (mife + miso and miso only) | 9 | 19.1 | -- | -- | 9 | 9.8 | 1 | 2.4 | -- | -- | 1 | 0.8 |
| Ayurvedic pills | -- | -- | -- | -- | -- | -- | 4 | 9.8 | 10 | 11.6 | 14 | 11.0 |
| Emergency contraception | 5 | 10.6 | 5 | 11.1 | 10 | 10.9 | 0 | 0.0 | 2 | 2.3 | 2 | 1.6 |
| Antibiotics/other pills | 6 | 12.8 | 3 | 6.7 | 9 | 9.8 | 1 | 2.4 | 0 | 0.0 | 1 | 0.8 |
| Traditional methods/other | 2 | 4.3 | 1 | 2.2 | 3 | 3.3 | 0 | 0 | 3 | 3.5 | 3 | 2.4 |
| **State** | | | | | | | | | | | | |
| State 1 | 13 | 27.7 | 16 | 35.6 | 29 | 31.5 | -- | -- | -- | -- | -- | -- |
| State 2 | 19 | 40.4 | 10 | 22.2 | 29 | 31.5 | -- | -- | -- | -- | -- | -- |
| State 3 | 15 | 31.9 | 19 | 42.2 | 34 | 37.0 | -- | -- | -- | -- | -- | -- |
| **Total** | 47 | 100.0 | 45 | 100.0 | 92 | 100.0 | 41 | 100.0 | 86 | 100.0 | 127 | 100.0 |

[1]Percentages add up to more than 100% due to dispensing of more than a single drug-type by some SDPs

Using corresponding survey data from 756 (Nigeria) and 1,762 (India) reproductive age female residents of the respective 47 and 43 EAs, over one-half (52.8% and 56.8%) resided in EAs with some availability of MA pills (S2 Table). In the multivariable adjusted models, women in Nigeria with secondary (odds ratio (OR): 10.94, 95% confidence interval (CI): 1.86–33.38) or higher education (OR: 17.61, 95% CI: 2.31, 48.05) and women in the second highest (OR: 15.83, 95% CI: 1.49–135.90) and highest wealth quintiles (OR: 15.29, 95% CI: 1.44–140.33) had significantly higher odds of living in an EA with availability of MA pills compared to women who never went to school and the poorest women (Table 3). In the Indian state, having any wealth compared to being in the poorest quintile, remained independently associated with MA pill availability in EAs (second poorest quintile: OR: 2.5, 95% CI: 1.07–5.58; wealthiest quintile: OR: 17.51, 95% CI: 3.64–57.47).

## Dispensing characteristics

Male clients were significantly less likely than female clients to be offered MA pills in Nigeria (27.2% vs 42.4%), whereas the opposite held true (42.2% vs 22.2%) in the Indian state (S3 Table). The MA drug regimens offered varied by country. Most dispensing SDPs in Nigeria (68.1%) offered the misoprostol-only regimen while a minority offered the combination regimen (12.8%) only, and about one in five facilities (19.1%) gave a choice of either regimen (Table 2). The median cost of the combined regimen was N4500 (US $12.20), significantly higher than the median cost of N1500 (US $4.07) for the misoprostol-only regimen. In the

**Table 3. Bivariate and multivariable regressions of characteristics associated with living in a community with medication abortion pills available in the local retail market among women aged 15–49 in three Nigerian states and one Indian state.**

| Characteristic | Nigerian states (n = 743) | | | | | | Indian state (n = 1,762) | | | | | |
|---|---|---|---|---|---|---|---|---|---|---|---|---|
| | OR | 95% CI | | aOR | 95% CI | | OR | 95% CI | | aOR | 95% CI | |
| **Age** | | | | | | | | | | | | |
| 15–19 | REF | - - | - - | REF | - - | - - | REF | - - | - - | REF | - - | - - |
| 20–24 | 1.19 | 0.72 | 1.98 | 1.11 | 0.64 | 1.91 | 0.98 | 0.75 | 1.29 | 0.88 | 0.63 | 1.22 |
| 25–29 | 1.56 | 0.92 | 2.65 | 1.33 | 0.78 | 2.27 | 1.12 | 0.81 | 1.55 | 1.23 | 0.78 | 1.94 |
| 30–34 | 1.32 | 0.80 | 2.20 | 1.17 | 0.69 | 2.00 | 1.03 | 0.67 | 1.57 | 1.13 | 0.64 | 2.00 |
| 35–39 | 1.13 | 0.72 | 1.80 | 1.23 | 0.80 | 1.90 | 1.02 | 0.71 | 1.46 | 1.17 | 0.67 | 2.03 |
| 40–44 | **1.76** | **1.05** | **2.95** | **2.28** | **1.14** | **4.59** | 1.21 | 0.77 | 1.91 | 1.42 | 0.75 | 2.71 |
| 45–49 | 1.18 | 0.60 | 2.29 | 1.54 | 0.71 | 3.35 | 1.17 | 0.71 | 1.91 | 1.38 | 0.64 | 2.94 |
| **Education** | | | | | | | | | | | | |
| Never | REF | - - | - - | REF | - - | - - | REF | - - | - - | REF | - - | - - |
| Primary | 4.98 | 0.91 | 27.30 | 3.38 | 0.98 | 11.69 | **1.53** | **1.09** | **2.14** | 1.35 | 0.89 | 2.05 |
| Secondary | **10.94** | **1.72** | **69.34** | **7.88** | **1.86** | **33.38** | **2.93** | **1.81** | **4.76** | **2.11** | **1.05** | **4.22** |
| Higher | **17.61** | **2.51** | **123.57** | **10.53** | **2.31** | **48.05** | **2.47** | **1.30** | **4.69** | 1.47 | 0.63 | 3.42 |
| **Wealth** | | | | | | | | | | | | |
| Poorest | REF | - - | - - | REF | - - | - - | REF | - - | - - | REF | - - | - - |
| Second Poorest | 8.53 | 0.92 | 79.25 | 7.63 | 0.96 | 60.52 | **2.50** | **1.12** | **5.59** | **2.44** | **1.07** | **5.58** |
| Middle | 9.15 | 0.85 | 98.30 | 7.83 | 0.83 | 73.59 | **4.94** | **1.69** | **14.39** | **4.41** | **1.50** | **12.98** |
| Second wealthiest | **15.83** | **1.45** | **173.17** | **14.21** | **1.49** | **135.90** | **6.89** | **2.10** | **22.55** | **5.90** | **1.71** | **20.37** |
| Wealthiest | **15.29** | **1.35** | **173.23** | **14.20** | **1.44** | **140.33** | **17.51** | **4.77** | **64.33** | **14.46** | **3.64** | **57.47** |

*Note*: Estimates in bold indicate statistical significance at the p<0.05 level

Model in Nigeria includes adjustment for state

OR = Odds Ratio; aOR = Adjusted Odds Ratio; CI = Confidence Interval, REF = Reference Category

Indian state, 82.9%, 14.6%, and 2.4% dispensed the combination regimen, the misoprostol-only regimen, or both regimens, respectively (Table 2). The median cost for the combination regimen at INR 600 (US $8.10) was slightly higher than INR 500 (US $6.75) for the single drug regimen. Among SDPs that offered to dispense some type of medicine, 39.3% in Nigeria and 33.3% in the Indian state, offered non-recommended products as abortifacients together with or instead of MA pills. These were most commonly emergency contraceptive and antibiotic pills in Nigeria, and ayurvedic pill preparations in the Indian state.

### Quality of care indicators

**Eligibility assessment and technical competence.** Among facilities offering MA pills in Nigeria, 61.7% and 70.2% respectively asked about pregnancy confirmation and gestational age information to determine eligibility (Fig 1); 75.6% and 56.1% asked for this information in the Indian state (Fig 1).

Considering prompted and unprompted responses, just over one-quarter (25.5%) of staff in Nigeria provided correct information on route of drug administration; over three- quarters (78.0%) did so in the Indian setting. Among staff offering the combination regimen, 46.7% in Nigeria and 88.6% in the Indian state provided accurate guidance on the order in which to take the pills; however, correct information on the appropriate interval between pills was low in both settings (26.7% and 14.3%).

Bleeding was the most discussed symptom with over three-quarters (76.6%) in Nigeria, and just over one-half (51.2%) providing this information in the Indian state; abdominal cramps

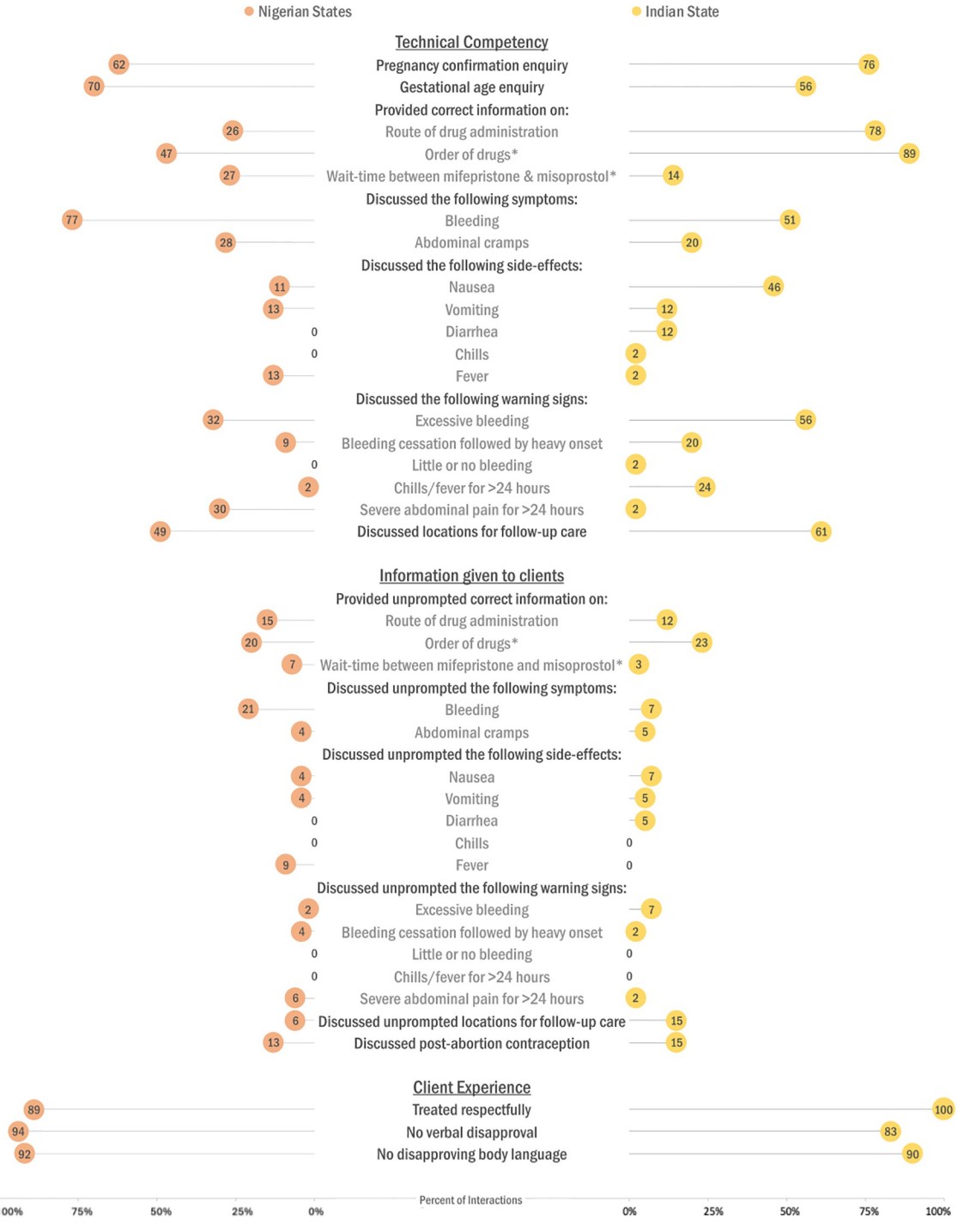

**Fig 1. Distribution of the quality of abortion care indicators by domain in the three Nigeria states (n = 47) and an Indian state (n = 41).** In both settings, performance was best on indicators related to client experience, with most room for improvement in the information given to clients domain. *Only if combination regimen offered; n = 15 facilities in Nigeria states, n = 35 facilities in the Indian state. NOTE: For technical competency, credit was given for correct information whether it was provided prompted or unprompted. For information given to clients, credit was only given for information provided unprompted.

were discussed in far fewer interactions in both settings. Discussion of side-effects was limited in Nigeria, e.g., only 12.8% discussed vomiting and fever, with no staff discussing potential for diarrhea or chills. Nausea was most discussed in the Indian state (46.3%), with other side-effects mentioned much less often. Excessive bleeding as a warning sign was discussed more frequently in the Indian compared to the Nigerian settings (56.1% versus 31.9%); other abnormal bleeding patterns were discussed to a lesser extent or not at all in both study locations.

**Information given to clients.** When accounting only for unprompted responses, accurate information provided to clients decreased considerably (Fig 1). In Nigeria, the relative reduction ranged from 33.3% for discussion of fever as a side-effect (12.8% to 8.5%) to 93.3% for discussion of excessive bleeding as a warning sign (31.9% to 2.1%). In the Indian setting, the decrease ranged from 60.0% for side-effects of vomiting and diarrhea (12.2% to 4.9%) to a 100% reduction for discussion of the side-effect of fever and little to no bleeding as a warning sign (2.4% to 0.0%).

**Client experience.** Client experience was positive in both settings. All clients in the Indian state and a majority (89.4%) in Nigeria reported being treated respectfully. A minority indicated that their request was met with some form of disapproval, which in the Indian state was more often verbal (17.1%) than body language-related (9.8%); 6.4% and 8.5% experienced these types of disapproval in Nigeria.

## Sensitivity analysis

As a sensitivity analysis with the Nigeria data, we examined changes in the distribution of individual indicators when we took a more conservative approach by recording an appropriate action only if it occurred in interactions with both the male and female simulated clients at a given facility. The absolute reduction in the percentage distribution of individual indicators was, on average, seven percentage points, and ranged from no change for nausea as a side-effect to a reduction of 23 percentage points for discussion of locations for follow-up care (S4 Table).

## Domain scores

Scores varied considerably across the three quality of care domains. The technical aspects of quality showed the largest deficiencies in both settings (Fig 2). The average score for technical competency was 18.4% (95% confidence interval (CI): 15.8–21.1) in Nigeria and 34.2% (95%

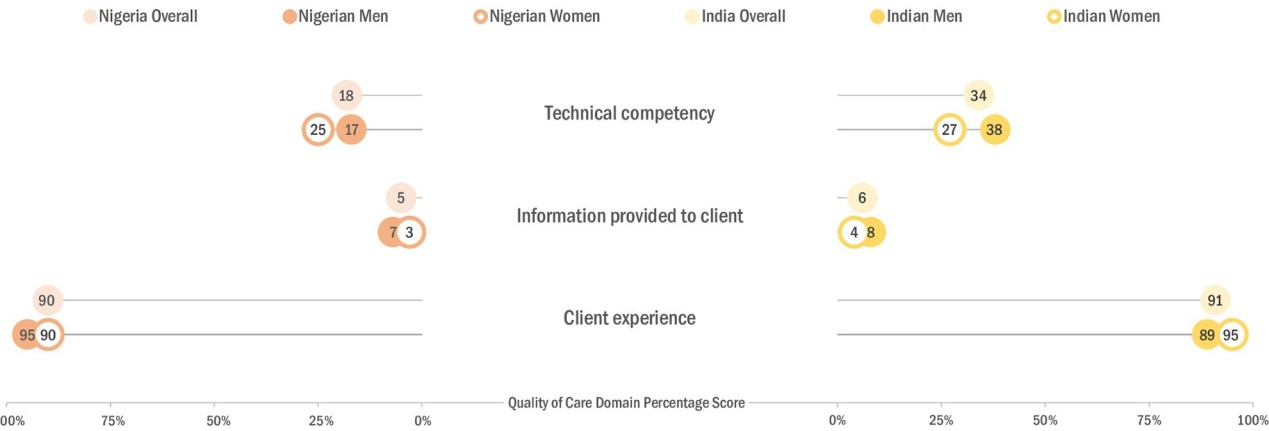

**Fig 2. Despite different legal contexts, quality of abortion care provided by pharmacies and chemist shops in Nigerian states (n = 47) and the Indian state (n = 41) were similar across domains.** Gender differences were not significant, but patterns differed.

CI: 29.1–39.2) in the Indian state. The corresponding mean domain scores for information given to clients were even lower at 4.5% (95% CI: 2.4–6.6) and 6.4% (95% CI: 3.2–9.7), respectively. The client experience domain had the highest mean scores at 90.1% in Nigeria (95% CI: 83.9–96.2) and 91.1% in the Indian state (95% CI:84.6–97.5).

Disaggregating mean domain scores by gender of simulated clients revealed different findings depending on the study context (Fig 2). The scores in Nigeria for technical competency were higher for female versus male clients (25.4% vs 17.2%), whereas the scores for information given to clients (7.2% vs 3.3%) and client experience (94.7% vs 89.7%) were higher for male versus female clients. In the Indian state, scores for technical competency (38.0% vs 26.8%) and client information (7.6% vs 4.2%) were higher for male clients, whereas client experience was higher for female clients (95.2% vs 88.9%). None of these differences reached statistical significance in either setting.

## Discussion

Our findings indicate that MA pill dispensing by pharmacies and chemist shops is common, varies by setting, and is of uniformly low technical quality. In the three Nigerian states, over half (51%) of visited facilities offered MA pills, most commonly the misoprostol-only regimen. This proportion is higher than that reported in a contemporaneous study conducted in the Lagos area (41%) using a similar simulated client approach [40]. In comparison to previous studies, these findings collectively imply that access to recommended MA pills is improving in urban environments, a positive development with potential to reduce unsafe abortion-related mortality in the country.

In the Indian state, a smaller percentage (32%) of facilities offered MA pills, frequently in the form of the combination regimen. Given national estimates indicating that nearly three-quarters of all abortions are self-managed outside facilities [35], we anticipated higher levels of informal MA drug provision. Our results are likely to be partially explained by regulatory actions undertaken in this state, including raids of retail shops by government officials, ostensibly to combat sex selective abortions, which have resulted in many chemists no longer stocking the combination regimen. A contemporaneous study on MA drug availability in chemist shops in this setting corroborates this explanation, with over three-quarters reporting legal and regulatory concerns as their primary reason for not stocking these pills [41]. With a history of gender-biased sex ratios in certain Indian states, the government's prioritizing of law enforcement against fetal sex determination has overshadowed the implementation of services in line with the legal framework on abortion. Such measures have given rise to both uncertainty about legal grounds for abortion and created a "chilling effect" on providers, including drug retailers, who fearing prosecution, may increasingly stop stocking the combination regimen.

We found evidence of some facilities in the Indian state providing ayurvedic pill preparations in addition to or instead of MA pills, a phenomenon previously documented [36, 37]. Similarly, in the Nigerian states, approximately one in five facilities offered to dispense non-recommended products. These data support findings from PMA's population-based female survey that approximately 10% and 8% of abortions in Nigeria and the Indian state respectively were conducted using non-recommended pills [42]. To the extent that these dispensing practices are shaped by the "tug" of punitive legal and policy environments in which providers operate while responding to the "pull" of social and financial incentives to meet ongoing demand, women in these settings are at risk of preventable abortion-related morbidity when non-recommended products are dispensed in substitution for MA pills.

We found variation in the distribution of MA drug availability by characteristics of reproductive-aged women living in the communities served by sampled facilities. Specifically, women with no education in Nigeria and poorest women in the Indian state were significantly less likely to have MA pills available at pharmacies and chemists in their communities. These findings suggest the persistent roles that structural factors play in shaping inequities in access to recommended abortion pills, even in the informal health care environment, and consequently in the risks and harms experienced across the spectrum of self-managed abortions. Such inequities may also inform method selection, with drug vendors deciding which pills to stock depending on the size and location of their outlets [36]. Even with the introduction of mifepristone into the retail market in Nigeria, the substantial price difference between the misoprostol-only and combined regimens will likely influence decisions among retailers and clients alike on the type of pills stocked or requested for self-managed termination. Cost and affordability play integral roles in influencing method choice as demonstrated by studies across settings [43–46]. While we did not systematically collect information on cost of ayurvedic pills in the Indian setting, information shared anecdotally by our simulated clients suggests that the cost of these pills is lower than the combination regimen, which may partly account for their popularity in the market. This assessment is consistent with a study of chemist shops in the Indian states of Bihar and Jharkhand where researchers found substantially higher demand and supply of ayurvedic pill preparations compared to MA pills, noting that this trend was likely driven by lower prices [36]. Variations in cost of pills are relevant for financing and equity considerations of self-care interventions as they become formally integrated into broader health system reforms [47]. Self-management of abortion in both these contexts relies entirely on self-financing as abortion remains a contested health service. Our findings suggest that in the absence of integrating modes of supply-side financing, existing inequities in the risk of unsafe abortion driven by dispensing practices that reduce out-of-pocket costs but involve the sale of ineffective or harmful products will likely persist.

We detected statistically significant differences in dispensing behaviors by client gender. Whereas in Nigeria female clients were more likely to be offered MA pills, male clients were more successful in the Indian setting. The latter finding is supported by previous research in India that has demonstrated men's central role in securing abortifacients on behalf of women [36, 48]. In both settings, we detected some differences in domain scores according to client gender, however these were not statistically significant. We recommend that future research investigate the drivers of differential dispensing practices in various contexts to better illuminate the role of gender and its intersection with other structural factors in influencing dispensing behaviors.

The technical aspects of quality associated with dispensing MA pills were particularly low, specifically in the domain of information given voluntarily to the client. These findings are consistent with previous literature indicating low adherence to evidence-based care in pharmacy-based dispensing of these pills [10]. The differences in scores between the technical competency and information given to client domains are an empirical demonstration of the know-do gap. In this case, the levels of knowledge are already low to begin with, as many of these providers are unlikely to have undergone training on medication abortion care, functioning as they do on the fringes of the regulatory system.

SDPs in Nigeria and the Indian state performed best in the domain of client experience. The high scores in this domain may be understood through the socially embedded practices of such drug vendors within communities, and the development of reciprocal relationships defined by a shared understanding of client needs and provider dependency on client patronage. Their popularity with clients for treatment of various conditions, including pregnancy termination, has been linked to clients' desire for care that they perceive as low-cost, accessible,

anonymous, and rapid [10]. A nuanced examination of factors that shape these provider-client interactions is salient for the study of quality, particularly the interpersonal aspects, as pharmacy providers occupy a growing role in the delivery of abortion care.

Our study findings have several additional policy, programmatic and research implications. The WHO's normative guidance on self-care interventions for sexual and reproductive health and their updated 2022 abortion care guidelines have clearly articulated a role for self-management of MA as a recommended service delivery strategy. Both in Nigeria and India, current legal frameworks governing delivery of abortion care criminalize self-management of abortion, making it impossible for health systems in either country to develop and implement policies that align with current public health guidance on abortion self-care. Our study reiterates the critical and growing role of pharmacy and chemist staff in facilitating access to misoprostol and mifepristone. If they continue to operate outside regulatory boundaries, evidence consistently suggests that technical quality of accompanying care will remain sub-standard. What works to improve dispensing behaviors is a question that remains to be answered as evaluations of training programs have demonstrated improvements in knowledge, but translation to behavior change has been mixed [49–52]. To take a patient-centered approach to abortion self-management, health systems in both countries will need to pivot away from clinical "provider-to-receiver" models of care to health promotion approaches that prioritize improving user knowledge and education. Efforts directed at raising the knowledge of users themselves, for instance with label inserts, are still in the nascent stages of development but appear to have potential [53]. Finally, accompaniment models of self-managed MA, where individuals receive evidence-based information from trained counsellors have been successful not only in bridging some critical gaps in abortion care access but also demonstrating high levels of safety and effectiveness [6, 54, 55]. Research is needed to generate data on the current reach of these services, and accompanying economic evaluations can help inform the allocation of financial and human resources to take successful interventions to scale.

## Study limitations and strengths

This study has limitations. The small samples of facilities offering MA pills renders low statistical power to conduct additional analyses, including whether observed differences in the quality of care delivered to male versus female clients are attributable to factors other than chance. We present study findings on quality at the SDP-level, whereas simulated clients obtained information from a single staff member during each interaction. To the extent that there is variability in the quality of care delivered in a single SDP by different staff or by the same staff on different occasions, as demonstrated by our sensitivity analysis in Nigeria, this approach may over- or under-estimate quality delivered to clients. Since our findings indicate uniformly low quality across all sampled SDPs, the sensitivity of our conclusions to this limitation is likely very low. Additionally, the study was not designed to collect data on client outcomes, thus we cannot link outcomes to the quality of care received.

In our analysis of the distribution of MA drug availability by women's characteristics, the outcome of interest was whether at least one SDP offered MA pills within an EA, a proxy variable for residential community. A limitation of this analysis is that women living on the periphery of an EA with no availability may have easy geographical access to a facility in a neighboring EA, or to a facility close to her workplace or other areas she frequents. The low precision of reported effect estimates as indicated by wide 95% confidence intervals is a consequence of small numbers.

Nevertheless, the study has several strengths. First is our use of the simulated client approach—a gold standard for measurement of process quality [56]. Given the restrictive legal

and/or regulatory environments within which these providers operate, we were able to empirically demonstrate that this method overcomes the significant under-reporting associated with survey-based approaches. The current WHO technical guidance provided a scientific basis for measure selection to which we also added complementary indicators on interpersonal care; these data could only be captured as part of the simulated client interaction. Our ability to collect empirical data in two geographically, legally, and culturally disparate settings demonstrated feasibility of using these measures more widely in the field.

## Conclusion

The growing role that pharmacies and chemist shops play in delivering abortion care in many settings offers women safe alternatives to the use of dangerous or ineffective methods. Our results demonstrate that MA pills are available in the retail market in the three Nigerian states and in the single Indian state involved in this study, albeit to different extents. However, the quality of care associated with dispensing these pills is sub-standard in both settings, with the implication that women who are buying and using pills purchased through the retail market are not given adequate information to follow recommended protocols for most effective self-use. Capitalizing on this pathway to safer abortion care, women's broader access to MA pills must be complemented with adequate guidance on medication use to facilitate the most favorable health outcomes. Doing so at scale requires changes in the structural and healthcare environments, including harmonizing laws on abortion with human rights standards, formalizing roles and capacities of pharmacy and chemist shop workers in the delivery of abortion care, developing models of healthcare financing that reduce user out-of-pocket costs, and allowing healthcare beneficiaries to operate as informed and educated autonomous agents in matters of their health and wellbeing.

## Supporting information

**S1 Table. Description of simulated client profiles.**
(DOCX)

**S2 Table. Proportion of women aged 15–49 residing in communities with medication abortion pills available in the local retail market in three Nigerian states and an Indian state.**
(DOCX)

**S3 Table. Dispensing of medication abortion pills by service delivery points (SDPs) disaggregated by gender of the simulated client in three Nigerian states and an Indian state.**
(DOCX)

**S4 Table. Sensitivity analysis of change in percentage distribution of quality of care indicators if using a more conservative approach (v2) to indicator calculation in Nigeria.**
(DOCX)

**S1 Checklist. Inclusivity in global research.**
(DOCX)

## Acknowledgments

We would like to acknowledge our study staff who played the roles of simulated clients and collected the data that forms the basis for this work. We are grateful to Collins Ifunanya Ozoadibe and Punit Soni for programming and testing the data collection forms and monitoring

quality and completeness of incoming data. We appreciate assistance from Rita Wiwa, Kshitiz Sisodia and Gargee Gopesh in coordinating logistics for training and data collection. Thanks also to Neisha Opper for her expertise in enhancing data visualization, and Maggie Flood for editorial input.

## Author Contributions

**Conceptualization:** Mridula Shankar, Elizabeth Omoluabi, Funmilola M. OlaOlorun, Anoop Khanna, Danish Ahmad, Caroline Moreau, Suzanne O. Bell.

**Formal analysis:** Mridula Shankar.

**Funding acquisition:** Suzanne O. Bell.

**Investigation:** Mridula Shankar, Elizabeth Omoluabi, Funmilola M. OlaOlorun, Anoop Khanna, Danish Ahmad.

**Methodology:** Mridula Shankar, Elizabeth Omoluabi, Funmilola M. OlaOlorun, Anoop Khanna, Danish Ahmad, Caroline Moreau, Suzanne O. Bell.

**Project administration:** Mridula Shankar, Elizabeth Omoluabi, Funmilola M. OlaOlorun, Anoop Khanna, Danish Ahmad.

**Supervision:** Caroline Moreau, Suzanne O. Bell.

**Writing – original draft:** Mridula Shankar.

**Writing – review & editing:** Mridula Shankar, Elizabeth Omoluabi, Funmilola M. OlaOlorun, Anoop Khanna, Danish Ahmad, Caroline Moreau, Suzanne O. Bell.

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
